# Exercise Effects on Health-Related Quality of Life (HRQOL), Muscular Function, Cardiorespiratory Function, and Body Composition in Smokers: A Narrative Review

**DOI:** 10.3390/ijerph20196813

**Published:** 2023-09-23

**Authors:** Nduduzo Msizi Shandu, Musa Lewis Mathunjwa, Ina Shaw, Brandon Stuwart Shaw

**Affiliations:** 1Department of Human Movement Science, Faculty of Science and Agriculture, University of Zululand, KwaDlangezwa 3886, South Africa; mathunjwam@unizulu.ac.za; 2School of Sport, Rehabilitation and Exercise Science, University of Essex, Colchester CO4 3SQ, UK; i.shaw@essex.ac.uk (I.S.); b.shaw@essex.ac.uk (B.S.S.)

**Keywords:** quality of life (HRQOL), smoking, cardiorespiratory fitness, exercise intervention, cognitive function, tobacco-attributable diseases

## Abstract

Smoking is amongst the leading global threats with high incidences of preventable premature mortality, morbidity, and various chronic diseases. The World Health Organization (WHO) proclaims a decrease in the prevalence of daily smoking in both males and females from 1980 to 2012, however, the number of regular smokers since then has exponentially increased. The low socio-income status individuals contribute greatly towards tobacco-attributable diseases due to limited access to healthcare systems, mostly in developing countries as compared to developed countries. Smoking affects the optimal functioning of the human body, which results in altered body system processes. Although a high intake of nicotine can lead to prolonged adherence and dependence on smoking, other factors, such as an individual’s level of health-related quality of life (HRQOL), stress, depression, and anxiety, can produce similar effects. Smoking has a wide impact on lifestyle factors, which explains the increase in the number of sedentary smokers with decreased health fitness levels and poor lifestyle conditions. Therefore, this study seeks to investigate the exercise effects on health-related quality of life (HRQOL), muscular function, cardiorespiratory function, and body composition in smokers. Concurrently, exercise as an intervention has been sourced as a rehabilitation strategy during smoking cessation programmes to restore the diminishing health components, however, a high rate of relapse occurs due to intolerable withdrawal symptoms.

## 1. Introduction

Despite extensive bodily knowledge of the harm triggered by smoking, it still remains one of the leading causes of preventable premature mortalities and morbidities worldwide [1]. The projected mortality rate attributable to tobacco smoking is estimated by the World Health Organization (WHO) to exceed eight million by the year 2030, with an annual death count of six million [2]. The WHO further estimated that by that date, 70% of the affected people will be from developing countries [1,2]. Over 20 million deaths have been reported between 1965 and 2014 from chronic conditions caused by direct smoking and/or exposure to second-hand smoke [1]. Moreover, smoking has been associated with a significant number of chronic mortality cases worldwide, specifically, 3,804,000 cases of lung disorders, 7,787,000 cases of cardiovascular and metabolic diseases, and 6,587,000 cases of smoking-related cancers have been recognized as high-prevalence conditions linked to smoking [3].

Globally, cigarette smokers total approximately 1.1 billion, with a more significant contribution from those with limited access to healthcare systems and classified as low socioeconomic individuals [2,4]. Further, such a contribution provides a viable explanation of that population’s gradual contribution to tobacco-attributable diseases such as lung cancer [4]. However, various studies report an increasing utilisation of tobacco products with a high relapse response among educated individuals and university students [5,6,7].

## 2. Smoking Effects on Health-Related Quality of Life, Muscular Function, Cardiovascular Function, Lung Function, and Body Composition

### 2.1. Smoking and Health-Related Quality of Life (HRQOL)

Excessive nicotine content in tobacco products is well known for its detrimental effects, including the development of physical and psychological dependence with continued cigarette use [8]. Friends who smoke, poor academic performance (i.e., low levels of motivation, curiosity, passion, and readiness to learn as well as an undesirable learning mentality), absence of parental support, high alcohol intake, pro-smoking attitudes, and low socioeconomic position are important variables that predict the transition to regular smoking [9]. Nevertheless, various factors such as stress, sadness, anxiety, and a person’s health-related quality of life (HRQOL) may also contribute to this change [10]. When compared to non-smokers, smokers’ HRQOL has been reported to differ [11,12]. Self-reported remarks about one’s bodily and mental health are included in the broad, multifaceted concept of HRQOL [13]. Physical health, social interactions, psychological state, and environmental factors are the four categories that make up HRQOL [14,15,16,17]. On that note, HRQOL comprises people’s mental and physical health, which is critical for developing and maintaining a healthy lifestyle [18]. Smoking and HRQOL have an unfavourable connection, with HRQOL indicating a dosage dependency [15].

In this aspect, smokers with elevated stress levels frequently consider themselves as less healthy, have a poor sense of self-worth, and are more likely to engage in harmful lifestyle choices such as increased alcohol consumption/binge drinking, unsafe sex, usage of drugs, and heavy cigarette use [19]. Furthermore, smoking and stress levels have been linked to a variety of adverse consequences, including depression and anxiety [10,19]. Consequently, sadness, emptiness, anhedonia, poor self-care, poor focus, loss of capacity to function, sleep difficulties, feelings of worthlessness, and recurrent suicidal ideation are common in smokers with depression [13]. Because cigarettes are commonly considered to have a relaxing effect, increased anxiety levels are frequently linked with smoking initiation [20]. Among smokers, the most frequent anxiety disorders are generalised anxiety disorder (GAD) and anxiety disorder caused by substances/drugs [21]. The symptoms of GAD include persistent restlessness, easy weariness, difficulty concentrating, muscle tension, and disrupted sleep [20]. This is commonly recognized in people under the age of 30, but it can also occur in youth or early adulthood [21]. Substance/drug-induced anxiety disorder manifests itself quickly after intoxication or withdrawal from a substance, or during drug exposure [22].

### 2.2. Smoking and Muscular Function

Research has demonstrated that smoking can alter skeletal muscle fibres and reduce the capability of oxidative enzymes, which results in skeletal muscle dysfunction [23]. Further, studies have reported that quadriceps muscles of smokers generate a lower maximal force capacity than non-smokers [24,25], with smoking causing biological impairments in the functioning of the mitochondria to grow in multiple sizes and numbers [25], which accounts for the mitochondrial density reduction. A principle of disuse in smokers has been applied extensively to understand the diminishing capabilities of skeletal muscle functioning [26]. Mechanisms enforced by smoking during muscle wasting involve protein degradation and/or reduced protein synthesis [24].

In general, muscle fatigue resistance is related to the oxidative capacity of muscles [23], with smokers having a muscle fibre type transition from oxidative type I fibres to more glycolytic type II fibres [24]. This slow-to-fast fibre type transition could also be caused by the recently mentioned disuse principle, not necessarily due to smoking-attributable changes [23,24,25,26]. However, greater percentages of smokers are sedentary individuals, which could account for the muscle atrophy [26]. An extended duration of the smoking lifestyle causes arterial vascular resistance and the vasodilatory response to increase, leading to a decreased blood flow and thus oxygen delivery to the active muscle to diminish and contributes to the increased muscle fatigability in smokers [27]. Most of these physiologic and physical changes caused by smoking lead to the prevalence of chronic diseases, which results in a reduced life expectancy [14,24].

### 2.3. Smoking and Cardiovascular Function

Smoking impairs the optimal functioning of the heart and blood circulation and increases the risk of coronary heart disease, myocardial infarctions and cerebrovascular accidents, and cerebrovascular and peripheral vascular diseases [28]. The carbon monoxide produced from cigarette smoke and nicotine can cause the heart to increase its function by increasing the number of beats per minute (bpm), which later becomes a public health concern [23]. Several studies indicate that at rest, young adult smokers have a heart rate (HR) of two to three bpm faster than young adult non-smokers’ resting HR [25,26,27,28]. This increased workload of the heart increases the early onset of myocardial infarctions and cerebrovascular accidents [24,25,26].

Smoking predisposes an individual to atherosclerosis, which increases the pressure exerted by the blood on the lumen of the blood vessels, which places an individual at an increased risk of developing hypertension, with increased blood pressure (BP) and mean arterial pressure (MAP) [24]. Smokers usually present with elevated myocardial workload levels with reduced exercise capacity and thus lower overall cardiovascular fitness [23]. Further, research has indicated that young male smokers have an elevated rate pressure product (RPP) and a reduced myocardial blood flow at rest compared to young male non-smokers [23,24,25,26].

A variety of research has shown that a number of changes in the autonomic nervous system in smokers is the outcome of the effects of nicotine and other substances that are found in cigarette smoke [26]. Elevated levels of nicotine mediate an increase in sympathetic nervous system functioning and release a cascade of hormones in abnormal concentrations, i.e., epinephrine, norepinephrine, and vasopressin hormones [24,25]. The autonomic imbalance usually elucidated in smokers is associated with the effect of nicotine-mediated stimulation of the autonomic ganglia and adrenal medulla [25], resulting in an increased discharge of cardiac sympathetic fibres [24]. The enhanced activity of cardiac sympathetic fibres due to smoking causes increases in HR, BP, and myocardial contractility by stimulating the β1 adrenergic receptor and increases the coronary vasomotor tone through acting on the α2 adrenoceptor [22,23,24,25]. The presence of nicotine in smokers causes an imbalance in sympathovagal activity because of an impairment in baroreceptor sensitivity, which is also known as a cardiovascular risk factor [12]. The increased RPP in smokers could detail the nicotine-induced sympathetic overactivity, resulting in an increased coronary vasomotor tone by acting on the α2 adrenoceptor [24]. Interestingly, smokers have mostly been found with an RPP of more than 100, indicating greater cardiovascular disease risks [25]. Recent research concluded that the short duration of smoking and smoking history of low pack years (PY) increases the myocardial workload and the risk of cardiovascular disease [26].

### 2.4. Smoking and Lung Function

The repetitive exposure to the toxic chemicals found in cigarette smoke has been linked to the impairment of the respiratory system that has reduced the functioning and structure of the lungs [27]. Smoking affects all components of the respiratory system simultaneously, since all the passages from the nose and sinuses down to the smallest airways of the lungs allow direct communication with one another [27]. During smoking, carbon monoxide (CO), being one of the toxic chemicals released, changes the molecular structure of haemoglobin by displacing oxygen [23,24,25,26]. The body’s immediate auto-response is to increase the number of red blood cells to compensate for and prevent the early onset of hypoxia [23]. Smoking causes immediate and rapid responses to the respiratory system, initially triggering bronchospasm [26]. Bronchospasm causes hypoxia in the body, creating breathing difficulties, while the body responds by increasing minute ventilation [27].

Further, cigarette smoke paralyses the cilia allowing mucus to increase in the lungs of a smoker, which increases phlegm production [16,26]. The chemicals and toxic substances entering the lungs are trapped by the mucus produced by the lungs [19]. Cigarette smoking further exacerbates mucus production, which stimulates goblet cell growth [11,23]. Due to the cilia’s paralysis, a cigarette smoker cannot clear the increased mucus [25]. This causes persistent coughing in an attempt to clear the lungs and bronchial passages [11]. Smoking-attributable diseases associated with this impaired respiratory system include pneumonia, lung cancer, and emphysema [28]. The WHO’s statistics reveal that 84% of deaths from lung cancer and 83% of deaths from chronic obstructive pulmonary disease (COPD) result from smoking [2]. The American Thoracic Society (ATS) outlined that the causes of COPD are associated with the period of exposure to tobacco environments and previous and current cigarette smoker’s genetic (inherited) risks [29]. Interestingly, a dose–response relationship exists between lung function and smoking, with a smoking reduction indicating improvements in lung function [23,24,25]. However, a non-linear relationship exists between smoking and cardiovascular disease. In this regard, recent research has noted that low levels of cigarette consumption can predispose a smoker to equivalent incidences of cardiovascular diseases when compared to heavy smokers (high nicotine intake) [13,26,29].

### 2.5. Smoking and Body Composition

Various studies have reported that smokers are underweight {lower body mass index (BMI)} when compared to non-smokers [10,19,30]. However, evidence suggests that smokers suffer from developing central fat accumulation, mostly seen among female smokers [30]. Since nicotine has a significant suppressing component, research has identified it as the reason smokers have a lower BMI than non-smokers [28,29,30]. This is because smoking increases the metabolic rate, decreases metabolic efficiency, and decreases caloric absorption and contrarily, results in cardiometabolic-related conditions [31].

The waist circumference, or the waist-to-hip ratio (WHR), is an indicator of the amount of visceral adipose tissue [24,29], with a high amount of visceral adipose tissue predisposing an individual to a higher risk of metabolic syndrome, diabetes, and cardiovascular disease [24]. Cross-sectional studies have identified smokers to have higher WHR’s when compared to non-smokers [22,28]. Further, a dose–response relationship exists between the WHR and the number of cigarettes smoked [32]. Possible mechanisms for greater waist circumferences have been strongly associated with cortisol concentrations in visceral adipose tissue [31]. Smokers have higher fasting plasma cortisol concentrations than non-smokers [24,31]. Further, research has indicated that smoking can result in fat storage by decreasing lipolysis [18,20,30]. This excess in fat storage can predispose a smoker to insulin resistance leading to a higher probability of diabetes among smokers [31]. Being overweight and obese demonstrates a well-established link between the increased risk of cardiovascular diseases [26,27,28,29,30,31] and the severity of cancers [26].

## 3. Exercise Effects on Health-Related Quality of Life, Muscular Function, Cardiovascular Function, Lung Function, Body Composition, and Smoking Cessation

The broad public health recommendation addresses the need to acquire a healthy lifestyle for smokers. Healthy adults should at least accumulate 30 min of moderate-intensity exercise on most, if not all, the days of the week [28], and individuals who are interested in enhanced outcomes exercised more often with a high-intensity [27,28,29,30,31]. Active research has noted a rise in the use of high-intensity interval training (HIIT) because one of the chief barriers usually foreseen when participating in exercise programmes is a lack of time [28,32]. HIIT involves repetitions of maximal and/or submaximal sprints for short and/or long periods, separated by recovery periods, between active or passive periods of rest [33]. HIIT benefits are observed in both healthy and immunocompromised individuals with alterations in processes of neuromuscular, molecular, metabolic, and cardiorespiratory functioning [33]. Interestingly, high-intensity interval running has been noted to be more enjoyable than moderate-intensity continuous exercise [31,32,33], with further notions of simultaneously producing adaptations in both anaerobic and aerobic exercise capacity [26], while continuous aerobic training (CAT) represents brisk walking, running, or cycling at a low-to-moderate intensity [30,31,32,33]. Traditionally, CAT, defined as exercise that is long in duration and primarily relies on aerobic energy metabolism, has long been the preferred form of training for most people [34].

### 3.1. Exercise and Health-Related Quality of Life

A dose-dependent relationship exists between exercise and the state of the HRQOL [35], with significant changes visible in the physical and mental health domains of the HRQOL during exercise [18,34]. Specifically, HIIT, other than CAT, has been reported to contribute positively towards HRQOL domains and has been found to reduce negative feelings of well-being [13,19]. On the contrary, resistance training causes positive results on the domains for about six months after the initiation of the exercise programme, then negatively contributes towards HRQOL [34,35]. High-intensity training has been reported as one of the effective methods of exercise to change the state of HRQOL in all populations, while low-intensity training has proven its effectiveness in changing the shape of the HRQOL in sedentary smokers [35]. The changes in the HRQOL are likely due to an increased adherence to the exercise programme due to social interactions, group participation, and time spent outdoors [32]. However, these benefits result from the stimulation of an increased release of endorphin levels and improved self-esteem, which involves positive self-reflection and body image [24]. Research evidence on exercise and HRQOL remains insufficient, as reported by recent studies, to evaluate different training intensities [35].

### 3.2. Exercise and Cardiovascular Function

During exercise, the heart of sedentary smokers is subjected to different intermittent haemodynamic stresses of pressure overload and volume overload [36]. The heart responds to the stresses of a higher demand of blood supply during recurrent exercise through morphological adaptation. This cardiac adaptation involves increasing the heart’s mass primarily by increasing the ventricular chamber wall thickness [16,27,32].

A recent study revealed the biomechanical changes that occur in smokers with a significant abnormality in the HR response [37]. In smokers during exercise, signals from the cardiac parasympathetic blockade cause the stimulation of mechanoreceptors to inhibit the stimulation of the activation of parasympathetic activity resulting in a drastic inverse outcome of its effects [27,35]. Exercise can reduce the cardiovascular effects of smoking by orchestrating the autonomic response to either increase or decrease the electrical stimulation of the pacemakers of the heart [26]. Further, exercise stimulates the autonomic control centre to increase the cardiac sympathetic activity and decrease the effectiveness of the cardiac parasympathetic nervous system, resulting in increased HR, stroke volume, and cardiac output and assisting in redistributing the blood flow to the active skeletal muscles [32,33,34,35]. Regular exercise training reduces the forces experienced by the arteries during cardiac muscle contraction, which lowers the BP and HR [26,27,28,29].

Studies have reported that cardiac sympathetic activity increases quickly at the onset of low-intensity aerobic exercise [28,33,36]. On the contrary, prolonged engagement in high-intensity exercise results in significant cardiovascular benefits such as low HR and BP during training and resting periods [34]. Household chores have been found to have a greater impact in reducing the effects of smoking, primarily due to weight maintenance, which further reduces the risks of early onset of cardiovascular disease [36]. Similarly, exercise can effectively be used to reduce or maintain body fat and thereby reducing the risk of developing cardiovascular disease [13,29,38].

Elevated levels of BP have been reported to respond positively to exercise interventions, with current evidence indicating a hypotensive effect of up to 22 h post-exercise among the general population [29,30,31,32]. Both HIIT and CAT are equally effective in reducing MAP and systolic blood pressure (SBP). Most studies report reductions in the mean value of eight mmHg of the resting SBP [32,33,34,35], with such improvements indicating an approximately 35% decrease in premature mortality resulting from unhealthy lifestyle practices [34]. Lower HR at rest and during exercise indicates an individual’s physical status, which shows an improvement in physical fitness in healthy individuals [36]. During exercise, stimulation effects experienced by the autonomic nervous system cause increases in sympathetic discharge [32], which significantly results in increases in HR, SBP, and RPP [32,33,34,35]. Further, these increases in HR, SBP, and RPP share a dose-dependent relationship with workload to provide an adequate blood supply during exercise to the active myocardium [24,35]. Increases are observed during participation in the HIIT type of training than with the long, slow continuous aerobic training, with HIIT progressing quicker towards the adaptation benefits [34].

Performing HIIT has been reported to improve mechanisms that contribute to enough oxygen delivery [13]. One efficient mechanism is the increase in vasodilation of the blood vessels that generally occurs during exercise, which reduces the viscosity and possibly increases the blood flow [26,28]. This is commonly seen with fast-twitch skeletal muscle fibres [36,37]. The blood flow increase resulting from a decrease in vessel resistance allows for rapid oxygen delivery to the functioning muscles, which increases the oxygen available for syphoning (extraction), which in turn may increase the maximum volume of oxygen consumption (VO_2_max) and time to the onset of fatigue [29,37]. Engagement in CAT causes cardiovascular adaptation, which includes increases in the mean peak power output, delay duration for the onset of fatigue, and VO_2_max [16,32]. CAT can also yield decreases in the time to completion of time trials and respiratory exchange ratio and reduce the resting HR levels [25,32]. These adaptations observed when performing a CAT usually require a long duration for optimal outcomes. The primary benefits of CAT are that it results in a significant increase in maximal oxygen uptake and increases oxygen delivery through a greater cardiac output, thus increasing the ability to perform work [38].

### 3.3. Exercise and Lung Function

Smoking causes elevated carbon monoxide levels in the blood, which reduces the ability of haemoglobin to carry oxygen to tissues and impairs the oxygen extraction by tissues [39]. Smoking causes ventilatory limitations, which result in an increased airway resistance and expiratory flow limitation leading to an increased breathing rate [40]. Smokers suffering from COPD have difficulty in performing their normal activities of daily living (ADL), physical activities, self-care, and hobbies [39,40], with symptoms that limit exercise, including leg fatigue, dyspnoea, and intermittent claudication [31]. This results in a sedentary lifestyle leading to deconditioning that further increases respiratory work related to any given task [37,38,39,40].

During exercise, severe COPD or airflow obstruction can lead to higher end-expiratory lung volume and impaired lung emptying, which worsens during the presence of any other negative risk factor (e.g., anxiety) [41], and this response of hyperinflation increases limitations of the tidal volume (Vt) response to exercise [39,40,41]. Moreover, COPD increases the elastic load on the inspiratory muscles and alters the muscle length–tension relationship forcing them into a shortened position [11]. The respiratory muscles are further limited in generating inspiratory pressures through electrolyte disturbances and steroid myopathy [42]. Limitations of respiratory ventilation in a sedentary smoker are being exacerbated by gas exchange abnormalities, which are being caused by excess physiologic dead space (Vd), a reduction in diffusion capacity, and the ratio between ventilation and cardiac output (VE/Q) mismatch [41]. Further, the early onset of lactic acidosis and hypoxemia caused by smoking leads to an increased ventilatory demand [43]. In smokers, exercise assists in prolonging or maintaining the life expectancy of a smoker by decreasing the risk of lung diseases and other associated diseases [41,42,43]. The ideal exercise training programme duration is yet unknown [43]. Research has extensively found significant gains during pulmonary rehabilitation using aerobic training [41,43]. Fewer studies have evaluated the effectiveness of HIIT on pulmonary rehabilitation [41,42,43].

Interestingly, both HIIT and CAT intensities can be individualized [43]. Moderate–high-intensity aerobic training has been associated with significant physiologic improvements in aerobic fitness through enhancing the activity of skeletal muscle oxidative enzymes [25,32]. The characteristics of physiologic changes showing improvements in aerobic fitness following exercise include a delay in the onset of anaerobic metabolism, increased muscle fibre capillarisation, mitochondrial density [26], and oxidative capacity of muscle fibres during exercise [43]. This reduces ventilatory requirements for a given exercise task leading to an increased VO_2_max and a decreased HR for a given oxygen consumption [41]. Respiratory muscle training (e.g., pursed lips breathing and diaphragmatic breathing) and lower limb and upper limb training have also been reported to provide significant benefits during pulmonary rehabilitation [41,42,43].

CAT has been shown to produce extensive pulmonary adaptations other than simple benefits such as respiratory muscle strength and endurance, commonly observed with HIIT [13,29]. A prolonged duration of exercise training increases the exerted workload on respiratory muscles, which results in increases in breathing rate and ventilation rates [44], which sets a higher demand for an increase in blood flow to the working respiratory muscles [44]. Various studies have reported a demand for blood flow of about 10 to 15% by the respiratory muscles during heavy exertion [39,40,41,42]. CAT predisposes an individual to the early onset of respiratory muscle fatigue due to the recruitment of additional respiratory muscles [44]. Little information has been documented on how respiratory muscles respond following whole-body exercise training, although the skeletal muscle adaptations to exercise training have been well documented [25]. The increase in respiratory muscle strength during endurance training has been related to changes in breathing patterns and active expiration through recruitment of additional respiratory muscles, which have been considered an underlying mechanism for greater respiratory muscle endurance [25,41,43].

Although studies to date have not elucidated enough evidence to support the belief that exercise training does not successfully alter the pulmonary system [43,44], HIIT has been recognised to lead to extensive alterations of the respiratory muscle strength and expiratory flow rates, which is not common when performing CAT [45]. The underlying mechanisms responsible for the acquired adaptations with HIIT have been associated with high ventilation and pressures [42]. A strong relationship exists between the cross-sectional area and the strength of a muscle, with evidence suggesting an immediate increase in the diaphragm thickness with HIIT training [41,42,43,44]. During training, the changes in breathing patterns adopted during HIIT could lead to increased flow rates through different underlying mechanisms within the pulmonary system [45].

### 3.4. Exercise and Body Composition

During exercise training, a significant decrease in the amount of fat-free mass, total body, and visceral fat occurs with an overall reduction of the whole-body mass [46]. Additional evidence has supported that significant changes are associated with the duration and intensity of training, with notable differences in body mass, BMI, waist circumference, fat-free mass, fat mass, and percentage of body fat, with rates of fat oxidation in grams per day increasing during training [31]. However, more favourable benefits are acquired earlier with, for example, HIIT [37]. A dose–response relationship between training volume and body-fat loss suggests that a higher amount of training leads to more significant body-weight reductions [47]. Early studies noted that fat metabolism is higher during moderate continuous training [39,40,41,42,43], with a recent study suggesting a higher total fat oxidation quantity after a HIIT session [26]. HIIT assists in improving the body composition by increasing body and skeletal muscle fatty acid oxidation and the inhibition of appetite and facilitating the release of corticotrophin hormone [37,46]. Moreover, there are some reasons to believe that the excess post-exercise oxygen consumption (EPOC) might assist in post-exercise fat metabolism [44].

Total energy expenditure has been reported as a critical factor in inducing fat loss [31]. Although these changes are observable during endurance training, it is less dominant and slower than when performing HIIT. Recent studies have taken note of endurance training individuals having a high energy expenditure, which could be caused by a sufficient EPOC [46,47]. Therefore, a linear relationship exists between the EPOC magnitude and exercise intensity in determining whether body composition and weight improvements will occur [48]. Since visceral fat is the major risk factor for cardiovascular diseases [48], a reduction through either HIIT or CAT is essential [13,29].

## 4. Conclusions

This narrative review highlights the multifaceted impact of exercise on various aspects of health-related quality of life (HRQOL), muscular function, cardiorespiratory function, and body composition in smokers. The evidence suggests that incorporating exercise into smoking cessation programmes can significantly improve HRQOL by addressing physical and psychological well-being. Moreover, exercise contributes to the restoration of muscular and cardiorespiratory function, counteracting some of the detrimental effects of smoking. While aerobic exercise can improve health fitness, psychological, and physical well-being extensively, there is a need to determine the effect of other modes of exercise training (i.e., resistance training) on these same variables since this would allow for programme design considerations that increase adherence and improve outcomes, possibly yielding even more significant health benefits in a shorter space of time and in resource-constrained environments [13,49]. In this regard, exercise-aided smoking cessation programmes with built-in maintenance components are well-known to enhance post-intervention cessation rates in the early weeks [50]. Moreover, exercise plays a pivotal role in enhancing body composition, including reducing body fat and promoting lean-muscle mass, which are essential for overall health [51]. However, it is important to note that the effectiveness of exercise as an adjunct to smoking cessation can vary depending on factors such as exercise type, intensity, and individual preferences [50]. In summary, while exercise offers promising benefits for smokers aiming to quit, a personalized approach that considers individual preferences and needs is crucial for success. Integrating exercise into smoking cessation programmes holds the potential to not only aid in quitting smoking but also to enhance overall health and well-being in the journey toward a smoke-free life. Further research and tailored interventions are warranted to optimize the positive effects of exercise in this context.

## Data Availability

Not applicable.

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
