# Peer review of "Exercise Effects on Health-Related Quality of Life (HRQOL), Muscular Function, Cardiorespiratory Function, and Body Composition in Smokers: A Narrative Review"

_ijerph, 2023, doi:10.3390/ijerph20196813_

Round 1
Reviewer 1 Report (Previous Reviewer 1)
The Authors have addressed all my concerns. I previously suggest the addition of a materials and methods section. This is not mandatory, however I think that may improve the manuscript. Anyway, if the Authors have intentionally chosen to not add this section, I would suggest answering in a more robust way to the Referee in order to understand the choice.
Author Response
Please see the attachment

Reviewer 2 Report (Previous Reviewer 3)
In this revised manuscript, the authors have attempted to address the concerns raised by the reviewers. However, despite their efforts, I remain unsatisfied with the revised version as the authors have not adequately addressed the critical points raised during the review process.
Of particular concern is the continued presence of improper citations. Although I will not list them separately, there are instances where previous studies are cited without relevance to the content. I cannot overlook these errors.
Author Response
Please see the attachment

Reviewer 3 Report (New Reviewer)
Thanks for putting up a narrative review on this broad topic and highlighting the positive effects of exercise.
1. Introduction
Line 32-33: Please clarify this statement; shouldn't mortality rate be the same as number of deaths?
Line 37-39: Please rephrase this sentence as it doesn't read quite right.
2. Smoking effects on HRQOL
Line 51: Should be 'low'
Line 136: Should be pack years
Line 144: Missing a full-stop after displacing oxygen
3. Exercise effects on HRQOL
Line 193: HIIT and CAT should be spelt in full on first mention
Line 199: low-intensity training
Line 247-248: Consider using 'enhanced' oxygen delivery
Line 253: VO2max should be spelt in full on first mention
Line 281: Explain ventilation-perfusion as the ratio between ventilation and cardiac output and thus VE/Q
Line 285: Delete 'the' before exercise training
Line 289: How can HIIT be administered at a low or moderate intensity? Please clarify this sentence.
Line 290: Are you referring to moderate- to high-intensity training? If so, it shouldn't be written as moderate-high-intensity aerobic training.
Line 343: EPOC is sufficient and not excess EPOC?
Lines 353 & 362: Why is it that at this late juncture HIIT and CAT are spelt in full?
4. Conclusions
Please beef up your conclusions to end the review with more impact!
Good
Author Response
Please see the attachment

Reviewer 4 Report (New Reviewer)
I do not see the conclusions supported by the research, it determines adherence issues, since the reading of the article does not address this issue.
You should update the bibliographic references, the most updated is from 2018. That will surely lead you to draw other conclusionsThe conclusions I do not see them supported by research, determines adherence issues, ccunado in the reading of the article does not talk about this issue.
You should update the bibliographic references, the most updated is from 2018. That will surely lead you to draw other conclusions
Round 2
Reviewer 2 Report (Previous Reviewer 3)
I am not satisfied with the submitted manuscript because the authors have not adequately addressed the most critical point raised during the review process, which is improper citations.
As an illustrative example, consider Reference 37. In lines 350-351, the authors suggested that "more favorable benefits are acquired earlier with, for example, HIIT [37]." However, Reference 37 does not mention HIIT anywhere.
These improper citations are still present in several sentences throughout the manuscript.
Author Response
Please see the attachment

Reviewer 4 Report (New Reviewer)
The corrections have been applied correctly.
Author Response
Please see the attachment

This manuscript is a resubmission of an earlier submission. The following is a list of the peer review reports and author responses from that submission.
Round 1
Reviewer 1 Report
Please, see the attached file.

Reviewer 2 Report
The paper is reviewing the literature on the effects of physical fitness on quality of life, cardiorespiratory function, and body composition in smokers.
Introduction: Effects of physical fitness is lacking in the introduction. it is only focusing on smoking and its effects on health. The reviewer suggests the authors to including more information about physical fitness and its benefits in the study variables.
The subheading on line 76.
This section was mainly describing the effects of smoking on muscles and its work. The sub-heading is too broad. Authors need to change the sub-headling or edit the section to represent actual physical fitness, not just muscle components.
Line101. The authors stated that smoking can cause the heart to increase its function by increasing HR. This sentence may confuse audiences thinking smoking has beneficial effects on the heart. The sentence needs to be rewritten.
Smoking and body composition
The first paragraph may mislead audiences into thinking smoking is a good thing for BMI and weight reduction. The reviewer is suggesting the authors edit this paragraph.
Exercise and body composition
The first reference #46 is incorrect. The reference is about HRV and different devices. It doesn't have anything to do with Ex training and body composition.
The authors need to check the citations carefully.
Reviewer 3 Report
In this review, the authors provide a comprehensive summary of the effects of smoking and exercise on physical fitness, HRQOL, cardiorespiratory function and body composition. Regrettably, the content of this manuscript significantly overlaps with numerous existing reviews in the field.
General comments
1. The research question of this review is unclear. The authors need to clarify it.
2. This review mainly consists of sections on the effect of smoking on HRQOL, physical fitness, cardiovascular and lung function, and body composition, as well as sections on the effects of exercise on these outcomes. However, there is significant overlap with existing reviews in the field, which reduces the novelty of this manuscript. Additionally, the title suggests a focus on the interaction between exercise and smoking, specifically the effects of exercise in smokers, but there is limited discussion on this topic.
3. Many references are improperly cited. For instance, ref. 16 should not cited in the text discussing HRQOL (line 54-55, page 2). I would like to believe these inaccuracies are merely typographical errors regarding reference numbers, but this raises concerns about the quality of the manuscript.
4. As review articles are often read by a wide audience, I strongly recommend adding figures or tables to enhance the understanding of the manuscript's contents.
Minor comment
5. I noticed several typos in the manuscript, such as in line 231-232. Additionally, some abbreviations are used before their definitions or are repeatedly defined.
